# CNS Redox Homeostasis and Dysfunction in Neurodegenerative Diseases

**DOI:** 10.3390/antiox11020405

**Published:** 2022-02-16

**Authors:** Gundars Goldsteins, Vili Hakosalo, Merja Jaronen, Meike Hedwig Keuters, Šárka Lehtonen, Jari Koistinaho

**Affiliations:** 1A.I.Virtanen Institute for Molecular Sciences, University of Eastern Finland, 70211 Kuopio, Finland; gundars.goldsteins@uef.fi (G.G.); vili.hakosalo@uef.fi (V.H.); merja.jaronen@uef.fi (M.J.); sarka.lehtonen@uef.fi (Š.L.); 2Neuroscience Center, University of Helsinki, 00014 Helsinki, Finland; meike.keuters@helsinki.fi

**Keywords:** mitochondria, reactive oxygen species (ROS), endoplasmic reticulum (ER) stress, unfolded protein response (UPR), neuroinflammation, ferroptosis, Alzheimer’s disease, Parkinson’s disease, amyotrophic lateral sclerosis

## Abstract

A single paragraph of about 200 words maximum. Neurodegenerative diseases (ND), such as Alzheimer’s disease, Parkinson’s disease, and amyotrophic lateral sclerosis, pose a global challenge in the aging population due to the lack of treatments for their cure. Despite various disease-specific clinical symptoms, ND have some fundamental common pathological mechanisms involving oxidative stress and neuroinflammation. The present review focuses on the major causes of central nervous system (CNS) redox homeostasis imbalance comprising mitochondrial dysfunction and endoplasmic reticulum (ER) stress. Mitochondrial disturbances, leading to reduced mitochondrial function and elevated reactive oxygen species (ROS) production, are thought to be a major contributor to the pathogenesis of ND. ER dysfunction has been implicated in ND in which protein misfolding evidently causes ER stress. The consequences of ER stress ranges from an increase in ROS production to altered calcium efflux and proinflammatory signaling in glial cells. Both pathological pathways have links to ferroptotic cell death, which has been implicated to play an important role in ND. Pharmacological targeting of these pathological pathways may help alleviate or slow down neurodegeneration.

## 1. Introduction

Redox homeostasis is acknowledged to play a major role in both normal cellular function and disease of the human central nervous system (CNS). This review aims to cover several major intracellular pathways affecting redox homeostasis and being relevant in the scope of degenerative diseases of the CNS. Here, we reveal several important linking axes between the pathways of mitochondrial reactive oxygen species (ROS) production and ferroptosis, and oxidative protein refolding in endoplasmic reticulum (ER) and neuroinflammation. Finally, we discuss the relevance of these pathways in the pathogenesis of several NDs, particularly Alzheimer’s disease, Parkinson’s disease, amyotrophic lateral sclerosis, and Huntington’s disease.

## 2. Redox Homeostasis in Mitochondria and Production of ROS

Mitochondria-dependent aerobic metabolism has a clear advantage for the production of energy-carrying substrates such as (adenosine 5′-triphosphate) ATP and maintaining body temperature or metabolite homeostasis. On the other hand, mitochondria also take the central position in the intracellular production of reactive oxygen species (ROS) [1]. Mitochondrial ROS generation is initiated upon electron escape from the electron transport chain (ETC) and the reaction of this electron directly with oxygen, resulting in one electron reduction and the formation of superoxide anion radical O_2_^−^ (Figure 1a). The consecutive reduction of superoxide gives rise to oxygen radical species and hydrogen peroxide, which is freely diffusible and relatively stable. The physiological superoxide detoxification mechanism is based on a dismutation reaction, which yields hydrogen peroxide and water molecules. This rection is catalyzed by superoxide dismutase (SOD) including cytosolic SOD1 and mitochondrial matrix SOD2. Hydrogen peroxide, in turn, is reduced by several enzymes comprising glutathione peroxidases (GPX), peroxiredoxins, and catalase. Even though primarily located in the cytosol, a fraction of SOD1 translocates to the mitochondrial intermembrane space (IMS). The role of SOD1 in this compartment is under debate, because IMS contains a high concentration of cytochrome C, which acts as an electron carrier in the ETC [2]. The oxidized form of cytochrome C has a prominent ability to oxidize superoxide, thus returning the escaped electron to ETC and preventing further ROS generation [3]. In contrast, an alternative pathway catalyzed by SOD1 in IMS produces hydrogen peroxide (H_2_O_2_), which may cause further cytochrome C oxidation and subsequent peroxidation of mitochondrial membrane phospholipids, including cardiolipin [4]. Accumulation of lipid hydroperoxides may promote ferroptosis, as discussed below.

To some extent, ROS are always generated during mitochondrial aerobic metabolism for intracellular signaling [5,6,7] However, redox activity imbalance may result in mitochondrial dysfunction. The excessive production of ROS contributes to oxidative stress and is a prominent part of aging processes and many degenerative diseases, including NDs [8]. Oxidative stress has been widely linked to the Aβ and α-synuclein proteinopathies in Alzheimer’s disease (AD) and Parkinson’s disease (PD), respectively. Previous studies have revealed that Aβ can induce mitochondrial dysfunction [9]. Damaged and dysfunctional mitochondria are extensively removed by mitophagy, which is a central mechanism for the maintenance of organelle homeostasis in neural cells [10]. Since neurons rely on oxidative phosphorylation as the main energy source, mitochondrial function is of utmost importance in studies relating to NDs. 

### 2.1. Alzheimer’s Disease

Alzheimer’s disease (AD) is the most prevalent ND with a multifactorial origin. The disease affects especially the cerebral cortex and hippocampus. The disease’s hallmarks are recognized as aggregates of beta-amyloid (Aβ) and tau proteins that cause the deposition of amyloid plaques and neurofibrillary tangles, respectively [11]. Aβ originates from the amyloid precursor protein (APP) that can undergo cleavage at different sites either by the non-amyloidogenic or amyloidogenic pathway, while tau aggregation is linked to its hyperphosphorylation [12].

Decades-long investigations have shown prominent mitochondrial abnormalities in the brains of AD patients [13]. Consequently, mitochondrial dysfunction is thought to be an early and prominent feature of AD, which is in concert with compromised energy metabolism observed to be evident already before disease onset [14]. These findings are based on fluoro-2-deoxyglucose PET imaging that indicates suppressed glucose utilization in the hippocampus and cortex of AD patients in comparison with individuals without dementia [15,16]. Consistent with this, disrupted mitochondrial bioenergetics and increased oxidative stress have been demonstrated in AD [17]. The mechanisms behind mitochondrial dysfunction in AD are thought to be related to impaired mitophagy/mitochondrial biogenesis, disturbed ER–mitochondria interaction, abnormal mitochondrial fusion, and fission and axonal transport deficits. Accumulation of swollen mitochondria with distorted cristae has been identified in both human AD cases and in transgenic animal models of AD [18,19]. On the other hand, peroxisome proliferator-activated receptor-gamma coactivator-1 alpha (PGC-1α), the master regulator of mitochondrial biogenesis, has reduced expression in AD patients and transgenic mouse models of AD [20]. In support of mitochondria-associated membrane disturbances, a recent study demonstrated that C99, the C-terminal fragment derived from APP by β-secretase cleavage, is present in mitochondria-associated ER membranes (MAMs). Importantly, the level of C99 is increased in AD resulting in reduced mitochondrial respiration [21]. In vitro overexpression of wild-type or mutant APP caused mitochondrial fragmentation in neuroblastoma cells and primary neurons [22]. The fragmentation observed was sensitive to beta-secretase 1 (BACE1) inhibition, suggesting a role of Aβ in excessive mitochondrial fission. Furthermore, inhibition of mitochondrial fragmentation with mitochondrial division inhibitor 1 decreased extracellular amyloid deposition, Aβ1-42/Aβ1-40 ratio, and prevented the development of cognitive deficits in a rodent model for AD [23].

Additionally, mitochondrial distribution is altered in the AD brain due to disturbed axonal transport. The mitochondria have a lower presence in the neuronal processes in the vulnerable AD pyramidal neurons when compared to healthy controls [24]. The abnormal distribution of mitochondria appears to be linked to ROS, since increasing acetylation levels of an antioxidant enzyme Prx1 by HDAC6 deacetylase inhibition recovered impaired mitochondrial axonal transport, decreased ROS levels and Ca^2+^ disbalance in AD models [25]. 

### 2.2. Parkinson’s Disease

Parkinson’s disease (PD) is the second most frequent neurodegenerative disorder with a multifactorial background. At the histopathological level, it is characterized by the presence of intracellular lesions named Lewy bodies and by exacerbated cell death of dopaminergic neurons [26]. Similar to AD, PD can arise due to genetic causes (familial or heritable) or occur sporadically [27].

PD belongs to synucleinopathies and is characterized by disrupted proteostasis of alpha-synuclein (α-Syn) that results from the formation of α-Syn oligomers and fibrils. In vitro investigations of α-Syn in a hypothalamic neuronal cell line GT1-7 revealed that its overexpression caused the formation of inclusion-like structures and mitochondrial deficits accompanied by increased levels of ROS [28]. More recent studies have demonstrated α-Syn translocation to the mitochondrial matrix and impairment of ETC complex I causing inhibition of ATP synthesis and a rise in ROS production [29]. Additionally, α-Syn oligomers reduced axonal mitochondria transport in induced pluripotent stem cell (iPSC)-derived neurons [30]. Moreover, disturbed mitophagy has been discovered in PD patient neurons, which was linked to abnormal accumulation of Miro protein [31]. Recently, the association of α-Syn with mitochondria resulted in decreased levels of mitochondrial SIRT3 and a reduction in mitochondrial biogenesis was demonstrated. This change was accompanied by impaired mitochondrial dynamics and significantly decreased oxygen consumption rate indicating mitochondria respiratory deficit [32]. 

### 2.3. Amyotrophic Lateral Sclerosis

Amyotrophic lateral sclerosis (ALS) is the most common progressive adult-onset motoneuron disease, characterized by selective death of upper and/or lower motoneurons. The disease leads to paralysis of voluntary muscles and eventually to death. The mechanisms causing neuronal degeneration in ALS are extensively studied, including disturbed motoneuron proteostasis [33]. While about 90% of ALS cases are sporadic, the remaining 10% are familial. Within familial ALS, the mutations in the SOD1 gene represent approx. 20% of dominantly inherited familial mutations, which cause the protein to misfold and aggregate. Another predominant familial cause for ALS, and also for frontotemporal dementia, is an intronic G4C2 hexanucleotide repeat expansion within the promoter region of chromosome 9 open reading frame 72-C9orf72 [34]. Even though the increased risk of ALS due to C9orf72 expansions is well documented, the exact mechanisms by which it causes neurodegeneration are not completely established [35,36]. Recent studies, however, indicate compromised mitochondrial function and bioenergetic deficits as key pathological factors in C9orf72 linked ALS [37,38].

Studies on animal models of familial forms of ALS have provided strong evidence for mitochondrial dysfunction in motor neuron degeneration [39,40,41]. Notably, there is also evidence of similar mitochondrial dysfunction in sporadic ALS cases. Analysis of patient autopsy samples has revealed increased mitochondrial density in motor neurons as well as significantly reduced ETC complex IV activity in the gray matter in both cervical and lumbar spinal cords [42]. Additionally, gene expression profiling in human induced pluripotent stem cells (iPSC)-derived motor neurons from sporadic ALS patients showed a strong association between mitochondrial functions and neurodegeneration [43,44]. Further studies on sporadic ALS patient iPSC-derived motor neurons have demonstrated that mitochondrial dysfunction in sporadic amyotrophic lateral sclerosis is developmentally regulated [45].

### 2.4. Huntington’s Disease

Huntington’s disease (HD) is an autosomal dominant ND with an average onset age of 40 years, characterized by expansion of polyglutamine repeats in the protein called huntingtin (htt). This leads to a gain of protein function toxic properties in the striatal and cortex neurons, causing motor and cognitive deficits [46]. More than 40 repeats predispose an individual to the disease, whereas less than 26 are tolerated [47]. There is a vast amount of research on HD models and patient samples indicating redox imbalance, as is evident from elevated levels of markers of protein oxidation and lipid peroxidation [48]. It has been demonstrated that mutant huntingtin (mhtt) affects the function of proteins and transcription factors related to mitochondrial homeostasis. It may cause inhibited expression of PGC-1α, a transcriptional coactivator that is known to regulate mitochondrial respiration and biogenesis [49]. Another line of evidence suggests compromised functioning of ETC manifesting as increased lactate levels and reduced activities of complex II, III, and IV in HD patients’ brains [50,51]. 

## 3. Link between ER Stress and Neuroinflammation

Intracellular protein accumulation is a hallmark of many NDs, including AD, PD, ALS, and HD [52]. Although the exact role of these protein aggregates in the pathology of neurodegeneration is under debate, one of their outcomes in the brain is endoplasmic reticulum (ER) stress.

Under physiological conditions, ER is responsible for protein synthesis, posttranslational processing, folding of newly synthesized proteins, and finally delivering the biologically active proteins to their proper target sites. Different harmful stimuli, for example, increased oxidation, neuronal aging, or mutations may result in the transcended activity of ER, and ER disruption in the normal physiological state, finally leading to ER stress. Recovery to its normal physiological balance requires activation of the unfolded protein response (UPR) signaling pathway. If the UPR fails to restore the cell integrity, cell death signaling cascades are activated and the cell undergoes apoptosis [53]. Furthermore, ER stress and UPR signaling are thought to contribute to the pathogenesis of NDs [54]. They can influence inflammation through various mechanisms, such as the production of ROS, the release of calcium from the ER, the activation of a key regulator of the inflammatory response transcription factor nuclear factor-κB (NF-κB) and the mitogen-activated protein kinase (MAPK) known as JNK (JUN N-terminal kinase), and the induction of the acute-phase response [55] (Figure 1b). In parallel, the ER stress-induced signaling pathway of PERK to translation initiation factor 2α (eIF2α) suppresses protein synthesis and promotes NF-κB-dependent transcription due to an increased ratio of NF-κB to IκB (inhibitor of nuclear factor kappa B) [56]. Moreover, upon ER stress, the interaction of IRE1α with TNF receptor-associated factor 2 (TRAF2) may lead to NF-kB activation [57,58]. 

In general, the mechanisms of ER stress contributing to the neuroinflammation and pathogenesis of particular NDs are not completely understood. Recently, the interplay between UPR and inflammatory pathways has been reviewed in more detail by Sprenkle et al. [59]. Depending on the disease type and the specific signaling molecule studied, manipulations of UPR in various disease models have yielded controversial and contrasting results [60]. A recent in vivo study has demonstrated that paraquat-induced ER stress caused a decrease in the number of neural stem cells and enhanced neuroinflammation in both subgranular and subventricular zones [61]. Besides aberrant inflammatory signaling described above, sustained ER stress compromises the protective function of the UPR in NDs, leading to the induction of apoptosis activating pathways and neuroinflammatory activation of glia. A growing body of evidence clearly indicates potential crosstalk between the UPR in immune cells and neuroinflammation [62].

### 3.1. Alzheimer’s Disease

Due to the protein aggregation seen during the disease progression, it is not surprising that several signs of ER stress have been demonstrated both in animal models and in human postmortem samples of AD [63]. Being a disease with multifactorial nature, AD displays concomitant synaptic loss with ER stress and neuroinflammation, which associate and correlate with the severity of pathology progression [64]. The research utilizing iPSC in deciphering the role of ER stress and UPR in AD seems to be scattered, possibly due to the immature or early developmental phase that iPSC-derived models typically represent. However, Kondo et al. have demonstrated ER and oxidative stress in iPSC-derived neurons both in familial and sporadic patients [65]. In addition, Oksanen et al. have demonstrated that the pro-inflammatory phenotype of presenilin-1 mutant iPSC-derived astrocytes is associated with elevated calcium leakage from the ER, which is a known hallmark of ER stress [66]. Interestingly, recent studies have discovered that ER stress can also stimulate the innate immune defense to protect the brain [59,67]. Despite the extensive work in studying the role of inflammation in AD, no clear answer to the question of how ER stress is linked to inflammation in AD has yet been achieved [68,69,70]. 

### 3.2. Parkinson’s Disease

A vast amount of research data has demonstrated the link between ER stress and UPR in PD pathology. Costa et al. [71] summarized the current evidence from post-mortem analysis and from in vivo and in vitro pharmacological and genetic studies. However, it is important to keep in mind that to some extent the evidence is controversial. Although inhibition of protein disulfide isomerase (PDI) is commonly seen as a beneficial molecule in ER stress, it can controversially suppress excessive protein folding and ER stress and induce clearance of aggregated α-Syn by autophagy as an alternative degradation pathway. These findings suggest a novel model explaining the contribution of ER dysfunction to MPP(+)-induced neurodegeneration, the most common toxin model of PD, highlighting PDI inhibitors as a potential treatment in diseases involving protein misfolding [72]. In parallel, ER stress induced in microglia and resulting in their neuroinflammatory phenotype has been implicated for the decrease in neural stem cells in an animal model of paraquat exposure [61].

### 3.3. Amyotrophic Lateral Sclerosis

The role of ER stress and UPR in ALS is widely studied by utilizing various disease models, such as ALS iPSC lines and animal models, as well as post-mortem ALS samples [54]. Interestingly, recent studies have demonstrated that misfolded protein accumulation increases PDI levels, promoting the cell death cascade [72,73]. In line with these findings, our lab has shown that UPR may lead to activation of PDI-dependent NADPH oxidase (NOX) and thus contribute to neurotoxicity in ALS [74]. In addition, lack of XBP1 (X-box-binding protein-1), a key UPR transcription factor that regulates genes involved in protein folding and quality control, has been shown to increase survival in an ALS mouse model [75] thus demonstrating the dual role of UPR in neurodegeneration.

### 3.4. Huntington’s Disease

Experimental evidence suggests that cytosolic mhtt protein fragments impair ER-associated protein degradation (ERAD) by entrapping ERAD proteins [76]. Consequently, misfolded proteins accumulate in the ER and cause ER stress. To counterbalance this accumulation, the UPR activates the proteosome degradation pathway and upregulates the transcription of chaperones [77]. Recent studies also suggested the interconnection between ER stress and UPR activation, leading to inflammatory processes [55]. Especially activated microglia and secreted inflammatory cytokines could lead to axonal damage and contribute thus to neuronal cell death in HD [78,79]. Overall, protein homeostasis maintenance is crucial for preventing mhtt-related toxicity, and modulation of the UPR pathway with specific inhibitors or activators could be the most effective therapeutic approach in the future.

## 4. Redox Imbalance and Protein Misfolding in Neurodegeneration

A vast proportion of translated proteins pass through the ER for tertiary modifications and ensure correct folding. Cells have an autoregulatory mechanism that detects and corrects misfolded or unfolded proteins that are sent to the ER [80]. Inside the ER, enzymes and chaperone proteins can help ER-imported proteins fold correctly by cleaving unfavorable intramolecular bonds or by helping to form new ones [80,81]. These modifications remarkably speed up the folding process, in which proteins fold into their native conformations. The ER also has a mechanism to detect a rise in the proportion of misfolded proteins in the ER, and when such an increase is detected, a downstream signaling pathway known as the UPR is launched to compensate for the need for more quality control proteins such as chaperones [82]. The overall role of UPR is to relieve a cell from ER stress and re-establish homeostasis. Three main routes become activated in the UPR, defined by different classes of transmembrane ER-resident signaling proteins. These three paths are mediated by the activating transcription factor 6 (ATF6), inositol-requiring enzyme 1 (IRE1), and the double-stranded RNA-activated protein kinase (PKR)-like ER kinase (PERK) [53]. UPR-activating transmembrane proteins are vastly studied in basic cellular biology and ER function, and they are prominent in the spectrum of protein misfolding-related NDs.

The levels of proteins inside cells are determined not only by synthesis but also by degradation rates, eliminating the consequences of faulty protein synthesis. Two major degradation pathways exist—the ubiquitin–proteosome pathway (UPP) and lysosomal proteolysis. In the UPP, the proteins for degradation are marked by ubiquitin attachment to the amino group of the side chain of a lysine residue [83]. After polyubiquitinated proteins are degraded in proteosomes, ubiquitin can be released and reused for another cycle. The clearance of misfolded proteins in NDs is primarily exercised by the UPP [84]. The second major pathway of protein degradation involves degradation in lysosomes. The proteins that are degraded by this pathway are long-lived cytoplasmic but dispensable proteins.

Under physiological conditions, all three UPR sensors are negatively regulated by the ER chaperone glucose-regulated protein 78/binding immunoglobulin protein (GRP78/BiP), which suppresses their activity by binding to their luminal ends [85]. When cells undergo constant ER stress, BiP dissociates from UPR sensors inducing their activation and thus promoting protein refolding and degradation of misfolded/unfolded proteins. However, under chronic ER stress, UPR sensors shift their signaling toward induction of cell death by apoptosis [81]. All three ER transmembrane proteins have an important control role in cell fate, where they can either help the cell survive in conditions that produce ER stress, or trigger apoptotic pathways when the cells sense that the pressure for extensive UPR is too high [86,87]. While the three ER proteins are perhaps the single most important molecules of the aftermath of ER stress, cells also have another mechanism to sense the ER stress state. These other mechanisms are pooled together in the non-canonical ER stress response, including ERAD, ER-related autophagy, and integrated stress response (ISR), respectively, which are thought to play an important role in NDs [88].

In the ER, redox balance is coupled to protein folding as well as cellular calcium homeostasis. Acute swings and prolonged changes in the redox balance can impair the cell’s capability to handle misfolded protein. The redox imbalance and protein misfolding can enhance each other and synergistically result in chronic ER-stress-related neurodegeneration [89].

Oxidation is used in the ER to help form a disulfide bond in an oxidative protein folding manner. The disulfide structures in a molecule are rearranged continuously by chaperone proteins such as PDI until the disulfides in a folding protein have obtained native conformation [90]. Oxidation of PDI is coupled with the reduction of Ero1 (Figure 1b). PDI can acts also as an isomerase for the disulfide bridges already present in the folding proteins, changing their conformation. The redox state of the CGHC motif of a PDI molecule determines whether the PDI acts as an oxidase or as an isomerase [74,91]. The levels of glutathione (GSH) indicate the oxidative balance of ER or cytosol and low levels of cytosolic GSH have been demonstrated in models of human NDs. The GSH and its oxidized form glutathione disulfide (GSSG) are present in the ER from 1:1 to 1:3 ratio (GSH/GSSG) compared to the >50:1 ratio present in the cytosol, due to the highly oxidative environment of the ER [92,93].

Protein misfolding, ER stress and redox state leading to neurodegeneration are common problems in a range of NDs, including AD, PD, ALS, and HD [94,95,96]. As reviewed in Hetz and Saxena (2017), maybe the most represented disease involving redox imbalance and protein aggregation is ALS [96].

### 4.1. Alzheimer’s Disease

Oxidative stress and redox imbalance, as shown by increased levels of lipid peroxidation end product 4-hydroxy-2-nonenal (4-HNE), are early events in AD pathogenesis [97,98] that precede amyloid plaque formation [99]. Accordingly, 4-HNE has been implicated to promote the formation of more toxic Aβ protofibrils [99]. Another piece of evidence for the role of oxidative stress and protein misfolding in AD is the finding that 4-HNE or Fe^2+^ can enhance the activity of β-secretase and thus increase the production of pathological Aβ42 [100]. This could be linked to the fact that Aβ40/42 can directly increase H_2_O_2_ production by metal transductors Fe^3+^ or Cu^2+^, creating a positive feedback loop in the presence of metal ions [101]. Metals such as Zn, Fe, and Cu have been observed to be enriched in amyloid plaques of AD patients and their significance has been reviewed elsewhere [102]. Interacting with metals is not the only way that Aβ has been shown to induce oxidative stress, as the soluble oligomers of Aβ and amyloid-beta precursor protein (APP) can inhibit the production of GSH by blocking the cysteine uptake through the EAAT3 receptor [103,104].

On the other hand, the generation of 4-HNE may induce an antioxidant response through activation of the Nrf2/Keap1 pathway [105]. It has been demonstrated that this pathway is particularly stimulated by the supplementation of docosahexaenoic acid (DHA) [106]. In concert with this, a number of recent studies indicate that long-chain polyunsaturated fatty acid-containing phospholipids are affected in AD, where depletion of DHA is the most prominent [107,108,109].

In parallel, studies on ER stress contribution to AD have demonstrated immunostaining against UPR markers such as pERK, elF2α, and IRE1α in the hippocampus. All these markers were correlated with intracellular Tau hyperphosphorylation in the hippocampal area, though surprisingly not with amyloid fibrils [110,111].

### 4.2. Parkinson’s Disease

In PD, intracellular fragmented compartments named Lewy bodies are a common finding and possibly the leading cause of PD, Lewy body dementia, and multiple system atrophy. Inside the Lewy bodies, the most prevalent protein is aggregated α-Syn, which can be found in a heterozygous pool of different-sized fibrils, intermediate species, and native monomers. The α-Syn is a small (14.5 kDa) protein with a fluctuating C-terminal and lipid membrane interacting middle-section and N-terminal. In a titration protein association assay, PDI was shown to bind monomeric α-Syn after 48 h incubation and with low dissociation constants with earlier time points [112]. In the same study, PDI was also shown to inhibit the fibrillization of α-Syn monomers. Further investigations showed that N-terminal residues V3-S9 and L38-V40, and residues at the C-terminus 123–127 and 135–137 of a wildtype α-Syn monomer interact with PDI [113]. They also confirmed inhibited fibrillization by PDI. Later, it was also shown that PDI can break down nascent α-Syn fibrils, yet not mature fibrils [114]. The redox balance in the ER impacts the S-nitrosylation state of PDI [115]. Interesting results have pointed out the importance of s-nitrosylation of PDI (SNO-PDI) towards α-Syn aggregation and Lewy body formation. In the PC12 cell line, treatment with a strong antioxidant ellagic acid prevented SNO-PDI and thus prevented the aggregation of α-Syn, synphilin-1, and α-Syn-synphilin-1 composites, also characterized as Lewy-like neurites [116]. Similar results have confirmed the importance of SNO-PDI for the aggregation of α-Syn [117]. The research revolving around protein folding regulation, PDI, and NDs is vivid and highlights the importance of oxidative status for such chaperone molecules that have dual activity. Even though the activity of PDI has been considered beneficial as it protects from protein misfolding accumulation, the excess activity of PDI can also be deleterious. A PDI redox state-coupled protein, Ero1a, can take an electron from oxygen, resulting in the formation of H_2_O_2_, an important oxidation producer in the ER. Ero1a can donate the electron along to PDI, resulting in the reduced form of PDI. On the other hand, Ero1a can receive the electron from PDI and form oxygen [118]. Our study demonstrated that inhibiting PDI or its redox regulator Ero1α exhibited neuroprotective properties in cell culture and *C. elegans* MPP+-induced toxicity models [72]. The inhibition of PDI or Ero1α by bacitracin or EN460, respectively, prevented MPP^+^-induced toxicity and α-Syn accumulation in the ER lumen. Moreover, it caused the clearance of aggregated α-Syn by autophagy. Similarly, treatment of iPSC-derived dopaminergic neurons carrying the GBA-N370S PD risk variant with tasquinimod, an allosteric inhibitor of HDAC4 or cantharidin, a protein phosphatase-2 inhibitor, reduced the α-Syn release, promoted autophagy, and ameliorated ER stress [119].

Like in AD, the presence of metals has been linked to protein misfolding and redox imbalance also in PD. The mechanism by which α-Syn oligomers induce ROS generation is related to redox metal ions and can be suppressed by the chelation of free iron or copper [120]. Moreover, it is known that metals also induce α-Syn aggregation [121]. Questions remain whether α-Syn interaction with metals causes fibrillization and then produces oxidative stress or whether α-Syn together with metal chelators first produces oxidative stress that promotes fibrillization, since there is the possibility that α-Syn fibrillization is a protective mechanism for toxic soluble oligomeric forms of α-Syn. All forms of α-Syn fibrils are known to interact with multiple different lipid formations and intracellular molecules. In the scope of oxidative stress findings where α-Syn is shown to interact with SOD1 and promote its oligomerization, it further links the α-Syn to an oxidative imbalance present in PD and in ALS [122].

### 4.3. Amyotrophic Lateral Sclerosis

In ALS studies, it has been demonstrated that the mutated SOD1 increases the expression of PDI in the ER and downregulates the ERAD pathway, indicating misfolding protein-induced ER stress [123]. Additionally, PDI upregulation can slow down the aggregation of SOD1 and dysfunctions of PDI, thus affecting the disease progression. In sporadic and familial ALS cases, PDI immunostaining correlated with the affected neurons [124]. Moreover, activation of UPR in both sporadic and familial forms of ALS has been demonstrated [123,125]. It is not thus surprising that clinical trials of a compound inhibiting eIF2α dephosphorylation for ALS treatment are ongoing [126,127]. elF2α is an important translation initiation factor and regulator of the integrated stress response.

### 4.4. Huntington’s Disease

While oxidative stress does not necessarily lead to protein misfolding or other pathological insults in HD since the disease is caused by genetics, the pathological protein huntingtin is involved in the misbalance of oxidative regulation and redox signaling. While the exact mechanism is still elusive, disruption of the antioxidant defense mechanism has been linked to HD pathophysiology [128]. In mice, treatment with mitochondria-targeted antioxidant MitoQ reduced markers of oxidative damage in muscle and significantly ameliorated fine motor control of R6/2 mice supporting the hypotheses that abnormal redox signaling in muscle contributes to altered proteostasis and motor impairment in HD [129].

## 5. Ferroptosis

Ferroptosis is a relatively recently discovered iron-dependent cell death mechanism, characterized by phospholipid peroxidation (Figure 1c). It is a regulated, non-apoptotic, intra-cellular cell death pathway [130,131]. Ferroptosis depends on the presence of intra-cellular free iron (Fe^2+^) and GSH-dependent activity of phospholipid hydroperoxide glutathione peroxidase 4 (GPX4) activity, which upon its reduced activity causes accumulation of lipid peroxides and initiation of ferroptotic cell death (Figure 1c).

The intracellular morphological hallmarks of ferroptosis include the shrinkage of mitochondria, an increased mitochondrial membrane density or rupture of the outer membrane, as well as a reduced number of mitochondrial cristae. Additionally, it is complemented by plasma membrane rupture, rounding of the cell due to cytoplasmic swelling, and cell volume loss [132].

Under physiological conditions, iron is taken up by mitochondria and ligated by heme into FeS clusters or to ferritin, a protein that stores iron intracellularly [133]. Nevertheless, heme catabolism also provides free iron (ferrous, Fe^2+^) for the Fenton reaction. Free and loosely bound intracellular iron that plays a central role in ferroptosis is assumed to enter the Fenton chemistry reactions, in which H_2_O_2_ is decomposed into harmful hydroxyl radicals. Fenton chemistry is a self-propagating chain reaction composed of the Haber–Weiss reaction and the Fenton reaction itself. In the Fenton reaction, Fe^2+^ reacting with H_2_O_2_ yields ferric iron (Fe^3+^) and hydroxyl radical (HO•). Next, hydroxyl radical reacts with H_2_O_2_ to produce superoxide (O_2_^−^). In the following Haber–Weiss reaction, superoxide reacts with H_2_O_2_ to produce HO• and hydroxyl anion (-OH). This is catalyzed by the reduction of Fe^3+^ to Fe^2+^, which in turn enters the Fenton reaction. The produced HO•s are strong initiators of lipid peroxidation [134,135,136]. These reactions can be quenched if enough radicals are formed so that they can react with each other by forming a bond and eliminating the radicals. Another way to stop the chain reaction is through antioxidative molecules [130,137]. The mechanisms above have high relevance for NDs such as AD and PD, where the accumulation of iron and oxidative stress have been shown to associate with degenerative pathology [138].

Besides the contribution of Fenton chemistry to lipid peroxidation, ROS/ reactive nitrogen species (RNS) interfere with proteins that regulate iron homeostasis through iron storage/release, thereby increasing the Fe^2+^ load. Fe^2+^ further interferes with the regeneration of endogenous GSH [134,135], hence GPX4 activity. The latter plays a central role in the protection of membrane lipids from peroxidation and is the most distinctly expressed GPX isoform in the brain [109,139]. By alternative splicing, cytoplasmic, nuclear, and mitochondrial isoenzymes of GPX4 are produced [139,140]. DHA plays an important role in GPX4 transcriptional regulation, and it has been demonstrated to induce GPX4 expression, particularly the cytoplasmic form, in brain cells [106,141].

Taken together, ferroptosis is mainly induced by the accumulation of Fe^2+^, depletion of GSH, and inhibition of GPX4 activity [142]. In the brain, iron overload may cause lipid peroxidation in neurons and glial cells, as well as GPX4 deficiency, as iron overload has been shown to be linked to motor neuron degeneration [143,144].

The relevance of ferroptotic cell death is further emphasized by its contribution to neuroinflammation. There is a multifaceted association between ferroptosis, arachidonic acid (AA) metabolism, and pro-inflammatory mediators [145]. The inflammation upon ferroptosis is triggered by the release of pro-inflammatory mediators and pro-inflammatory polarization of microglia/macrophages, the release of damage-associated molecular patterns (DAMPs), and immunogenic lipid metabolites [146,147,148]. Several pro-inflammatory factors released, such as tumor necrosis factor-alpha (TNF-α) and interleukin-1β (IL-1β), may provide positive feedback to ferroptosis by mediating neuronal iron uptake followed by sustained down-regulation of GPX4 activity [149,150]. In concert with this, inhibition of ferroptosis has been shown to decrease microglial activation and suppress the release of IL-6, IL-1β, and TNF-α [145].

Recently, ferroptosis has gained a lot of attention as a mechanism behind brain diseases, including NDs. Importantly, in NDs, ferroptosis appears to be driven by redox imbalance [151,152], and the intracellular pathways involved in ferroptosis allow to distinguish various druggable targets [153].

### 5.1. Alzheimer’s Disease

Despite characteristic pathological hallmarks of AD, including Aβ plaque and neurofibrillary tangles, the mechanism of neurodegeneration remains largely obscure. Since phospholipid peroxidation is apparent in AD patient brain samples, a growing body of evidence indicates the involvement of ferroptotic mechanisms in the pathogenesis of AD [154]. Elevated iron concentration was found in clinic-pathological examinations of AD cases many decades ago and targeting iron has been proposed as a disease-modifying therapy for AD [155]. On the other hand, there have been studies demonstrating the binding of iron to Aβ and tau while promoting their aggregation [156]. Since then, a meta-analysis of 300 AD cases from 19 studies has indicated the heterogenous distribution of elevated iron levels in AD brain cortical areas, particularly in the putamen and amygdala [157]. Moreover, hallmarks of eventual ferroptosis as GSH depletion, lipid peroxidation, and protein carbonyls have been found in post-mortem AD brain samples [158,159].

So far, several mechanisms have been proposed for linking iron with AD pathophysiology. First off, amyloidogenic processing of APP may destabilize ferroportin, a protein responsible for iron efflux, thus leading to elevated neuronal iron burden [160]. Next, pathological interaction between Aβ and iron may result in aberrant iron redox chemistry, leading to oxidative stress and cognitive deficits in AD [156]. In parallel, iron can promote tau hyperphosphorylation by induction of glycogen synthase kinase 3 beta (GSK3β) and cyclin-dependent kinase 5 (Cdk5) [161]. Finally, a correlation between enhanced neuronal default mode network activity in apolipoprotein E4 (APOE4) carriers and cortical iron burden has been demonstrated, suggesting that the interaction between APOE4 and iron may result in affected brain functions [162].

### 5.2. Parkinson’s Disease

In PD, several pathological hallmarks are consistent with ferroptosis [163]. Among them, hypermethylation-associated downregulation of cystine-glutamate antiporter SLC7A11 gene [164], elevation of lipid peroxidation products [165], increased iron concentrations and decreased GPX4 activity in *substantia nigra* [166,167], and DJ-1 depletion rendering neurons susceptible to the ferroptosis [168] are evident in PD models and are clearly linked to PD pathogenesis. Importantly, moderate iron chelation has shown promising therapeutical effects in PD patients [169]. A functional link between ferroptosis and PD may be related to disturbed α-Syn proteostasis, which is associated with iron and lipid metabolism [163].

### 5.3. Amyotrophic Lateral Sclerosis

Markers for oxidative stress and lipid peroxidation, such as malondialdehyde, 4-HNE, protein carbonyls, and oxidized membrane phospholipids, have been found in animal models and in samples from familial and sporadic ALS cases [170,171]. Importantly, deficiency in ferroptosis-suppressing GPX4 in neurons causes motor neuron degeneration and paralysis [143]. In contrast, overexpression of GPX4 in mutant SOD1G93A ALS model mice slows down disease progression [172]. Moreover, investigations on ferroptosis inhibitors, such as CuII(atsm), have recently shown positive results for phase 1 studies in ALS patients [173].

### 5.4. Huntington’s Disease

There is emerging evidence regarding the contribution of ferroptosis to HD pathology [174]. Most importantly, MRI imaging of HD patients has revealed increased iron deposition in several brain regions [175], and ferroptosis inhibitors, such as ferrostatin-1, exert positive effects on animal models of HD [176]. The mechanism of how the expression of mhtt is linked to the induction of ferroptosis may include interactions with the outer membrane of mitochondria and disturbances of calcium homeostasis [177], augmented mitochondrial fragmentation [178], and disturbed import of mitochondrial proteins [179]. Additionally, mhtt may interfere with iron endocytosis, thus leading to its increased accumulation [180].

## 6. Conclusions

CNS redox imbalance originating from mitochondrial and/or ER sources affects neural cell function, both in terms of viability and pro-inflammatory reactivity. As an ultimate outcome, it may trigger ferroptotic cell death. Since being recognized as having high relevance for several NDs, ferroptosis has attracted great interest as an opportunity to develop treatments for incurable NDs, such as AD, PD ALS, and HD. There have already been considerable efforts to develop small molecule ferroptosis inhibitors, particularly indicated for ALS, PD, and AD [137,173,181].

## Figures and Tables

**Figure 1 antioxidants-11-00405-f001:**
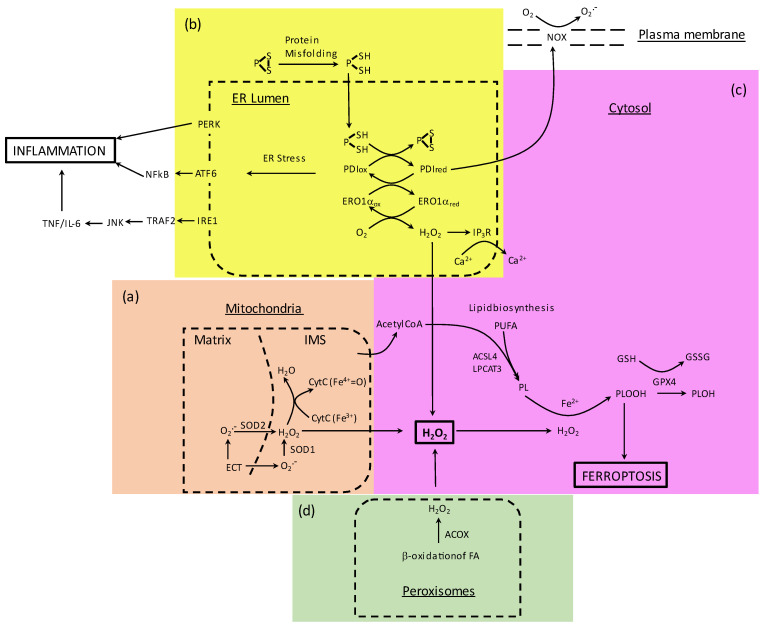
Major sources of intracellular reactive oxygen species (ROS) production and their convergence to ferroptosis through generation of H_2_O_2_. (**a**) In the mitochondria, ETC-produced superoxide is converted to hydrogen peroxide by SOD2 and SOD1. Cytochrome C, located in the mitochondrial IMS, may have a dual role by scavenging superoxide or interacting with peroxide and contributing to the cardiolipin peroxidation. Mitochondrial activity by supporting lipid biosynthesis may enhance ferroptosis. (**b**) Excessive protein refolding in ER increases PDI/Ero1a activities resulting in a rise of H_2_O_2_ production, calcium efflux, NADPH oxidase activation, and general ER stress linked to inflammation. (**c**) Ferrous iron overload, elevated levels of H_2_O_2_, and GPX4 deficiency cause accumulation of lipid peroxides and initiation of ferroptotic cell death. ACSL4 and LPCAT3 catalyze incorporation of PUFA into membranes, thereby sensitizing them to ferroptosis. (**d**) In peroxisomes, ROS are primarily produced by fatty acid beta-oxidation catalyzed by ACOX. Abbreviations: IMS—intermembrane space; ETC—electron transport chain; SOD1—Cu,Zn superoxide dismutase; SOD2—Mn superoxide dismutase; PDI—protein disulfide isomerase; ERO1a—Endoplasmic reticulum oxidoreductase 1 alpha; NOX—NADPH oxidase; GPX4—Glutathione peroxidase 4; PERK—protein kinase RNA-like endoplasmic reticulum kinase; ATF6—activating transcription factor 6; IRE1—inositol-requiring enzyme 1; NFkB—nuclear factor kappa-light-chain-enhancer of activated B cells; TRAF2—TNF receptor associated factor 2; JNK—c-Jun N-terminal kinase. ACSL4—Acyl-CoA synthetase long chain family member 4; LPCAT3—Lysophosphatidylcholine acyltransferase 3; PUFA—polyunsaturated fatty acids; ACOX—Acyl-CoA oxidase.

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
