# Peer review of "CNS Redox Homeostasis and Dysfunction in Neurodegenerative Diseases"

_antioxidants, 2022, doi:10.3390/antiox11020405_

Round 1

Reviewer 1 Report

The manuscript “CNS redox homeostasis and dysfunction in neurodegenerative diseases” by Goldsteins et al is a review article which describes the pathological mechanisms of neurodegenerative diseases including Alzheimer’s disease, Parkinson’s disease and Amyotrophic Lateral Sclerosis. In this review, the authors especially focus on imbalance of CNS redox homeostasis caused by mitochondrial dysfunction and endoplasmic reticulum stress. Generally, the subject is of interest and scientifically sound and contains essential issues. This topic is also of importance for treatment of neurodegenerative diseases. The manuscript has been well organized and written. However, I have some concern on the paper.

In this review, the authors mainly described the pathological mechanisms of neurodegenerative diseases such as Alzheimer’s disease, Parkinson’s disease and Amyotrophic Lateral Sclerosis. Please add the explanation why the authors discussed the above three neurological disorders in this review!

Also, there are other neurodegenerative diseases such as multiple sclerosis and Huntington's disease. Please add the description other neurodegenerative diseases, if possible.

The authors concisely discussed the imbalance of CNS redox homeostasis caused by mitochondrial dysfunction and endoplasmic reticulum stress, but a figure or table summarizing the redox homeostasis may be helpful.

Author Response

The manuscript “CNS redox homeostasis and dysfunction in neurodegenerative diseases” by Goldsteins et al is a review article which describes the pathological mechanisms of neurodegenerative diseases including Alzheimer’s disease, Parkinson’s disease and Amyotrophic Lateral Sclerosis. In this review, the authors especially focus on imbalance of CNS redox homeostasis caused by mitochondrial dysfunction and endoplasmic reticulum stress. Generally, the subject is of interest and scientifically sound and contains essential issues. This topic is also of importance for treatment of neurodegenerative diseases. The manuscript has been well organized and written. However, I have some concern on the paper.

In this review, the authors mainly described the pathological mechanisms of neurodegenerative diseases such as Alzheimer’s disease, Parkinson’s disease and Amyotrophic Lateral Sclerosis.

  1. Please add the explanation why the authors discussed the above three neurological disorders in this review!

Response: In the scope of the current review was to discuss the CNS redox homeostasis dysfunction common for the major neurodegenerative diseases including AD, PD, ALS and HD.

  1. Also, there are other neurodegenerative diseases such as multiple sclerosis and Huntington's disease. Please add the description other neurodegenerative diseases, if possible.

Response: The discussion of mechanisms described in Huntington’s disease has been added.

  1. The authors concisely discussed the imbalance of CNS redox homeostasis caused by mitochondrial dysfunction and endoplasmic reticulum stress, but a figure or table summarizing the redox homeostasis may be helpful.

Response: The figure included summarizes the major sources of intracellular ROS production and the mechanisms causing redox imbalance in different intracellular compartments. An additional table would basically duplicate the description provided in the figure and legend.

Reviewer 2 Report

MS# CNS redox homeostasis and dysfunction in neurodegenerative diseases

The manuscript by Goldsteins et al. reviews the complexity of redox deregulation in main neurodegenerative diseases (ND). The review is well structured and elegantly written, and integrates the present knowledge on the involvement of mitochondrial and endoplasmic reticulum oxidative stresses in the pathophysiology of ND through induction ferroptosis and inflammation.  Most information compiled in the manuscript have been solidly demonstrated in previous publications

I have only few comments, some of them related to relevant, yet missing, information

Page 2, Figure 1: Is it conceivable a oxidized for of iron to Fe+4 (in mitochondria)

Page, 3 Footnote of Fig. 1 should be placed just after the figure

Page, 8. Alzheimer´s disease paragraph. This section is poorly discussed and out of date. This has to be rewritten. Oxidative stress in AD has been extensively reviewed and current knowledge extends well beyond generation of 4-HNE from arachidonic acid, which, on the other hand, is substantially buffered by gluthathione (and thioredoxin) antioxidant systems in neural cells under highly oxidant environment (yet physiological) of brain parenchima. The imbalance between ROS/RNS generation (and concomitantly lipid peroxides) and maximal antioxidant systems capacity displaces the fine equilibrium between oxidative signalling and stress. In fact, other polyunsaturated fatty acids, in particular DHA, which is even more relevant for neuronal membrane function than arachidonic acid, is a source for HHE, a lipoperoxide by-product which, in response to moderate oxidative conditions activates the transcriptional activation of antioxidant systems, in particular GPX4 family of proteins, through Nrf2/keap1 pathway . Hence, lipoperoxide accumulation in nerve cells is not necessarily cause for oxidative stress. An excellent review on this matter has been published recently in this same journal.

Page 10. Given the key role of GPX4, the transcriptional regulation of GPX4 family of isoforms by indirect antioxidants must be discussed in this section.

Page 11, lines 451-453. It should be indicated that iron accumulation in post-mortem AD brains is heterogeneous, even between cortical areas, and not widespread throughout the brain. Some areas exhibiting iron overload are clearly affected in AD while others are mostly unrelated (Putamen, amygdala…). Whatever the reasons for this dissymmetry between brain areas are, it precludes a causative involvement of iron in.AD.

Page 11, lines 453-454.  GSH depletion, lipid peroxidation and protein carbonyls are not specific for ferroptosis

Page 11, lines 475-476. Check for abbreviations. 4-hydroxyl-2-nonenal  is used here expanded, but used several times as before as HNE without explained chemical name.

Author Response

Reviewer #2

The manuscript by Goldsteins et al. reviews the complexity of redox deregulation in main neurodegenerative diseases (ND). The review is well structured and elegantly written, and integrates the present knowledge on the involvement of mitochondrial and endoplasmic reticulum oxidative stresses in the pathophysiology of ND through induction ferroptosis and inflammation.  Most information compiled in the manuscript have been solidly demonstrated in previous publications

I have only few comments, some of them related to relevant, yet missing, information

  1. Page 2, Figure 1: Is it conceivable a oxidized for of iron to Fe+4 (in mitochondria)

Response: There is an evidence of formation of so called compound I-type species of cytochrome c cyt·+c−Fe(IV)=O. See for instance https://doi.org/10.1074/jbc.M300054200

  1. Page, 3 Footnote of Fig. 1 should be placed just after the figure

Response: Corrected

  1. Page, 8. Alzheimer´s disease paragraph. This section is poorly discussed and out of date. This has to be rewritten. Oxidative stress in AD has been extensively reviewed and current knowledge extends well beyond generation of 4-HNE from arachidonic acid, which, on the other hand, is substantially buffered by gluthathione (and thioredoxin) antioxidant systems in neural cells under highly oxidant environment (yet physiological) of brain parenchima. The imbalance between ROS/RNS generation (and concomitantly lipid peroxides) and maximal antioxidant systems capacity displaces the fine equilibrium between oxidative signalling and stress. In fact, other polyunsaturated fatty acids, in particular DHA, which is even more relevant for neuronal membrane function than arachidonic acid, is a source for HHE, a lipoperoxide by-product which, in response to moderate oxidative conditions activates the transcriptional activation of antioxidant systems, in particular GPX4 family of proteins, through Nrf2/keap1 pathway . Hence, lipoperoxide accumulation in nerve cells is not necessarily cause for oxidative stress. An excellent review on this matter has been published recently in this same journal.

Response: Additional text discussing mechanisms suggested has been included in the corresponding paragraph.

  1. Page 10. Given the key role of GPX4, the transcriptional regulation of GPX4 family of isoforms by indirect antioxidants must be discussed in this section.

Response: Additional text discussing mechanisms suggested has been included in the corresponding paragraph.

  1. Page 11, lines 451-453. It should be indicated that iron accumulation in post-mortem AD brains is heterogeneous, even between cortical areas, and not widespread throughout the brain. Some areas exhibiting iron overload are clearly affected in AD while others are mostly unrelated (Putamen, amygdala…). Whatever the reasons for this dissymmetry between brain areas are, it precludes a causative involvement of iron in.AD.

Response: The text corrected according to the suggestion.

  1. Page 11, lines 453-454. GSH depletion, lipid peroxidation and protein carbonyls are not specific for ferroptosis

Response: The text corrected according to the suggestion.

  1. Page 11, lines 475-476. Check for abbreviations. 4-hydroxyl-2-nonenal is used here expanded, but used several times as before as HNE without explained chemical name.

Response: Corresponding corrections introduced.

Reviewer 3 Report

The paper entitled “CNS redox homeostasis and dysfunction in neurodegenerative  diseases “by Ryszard Gundars Goldsteins et al. is a review paper within the scope of Antioxidants.  The present review focuses on the major causes of CNS redox homeostasis imbalance comprising mitochondrial dysfunction and endoplasmic reticulum (ER) stress. Mitochondrial disturbances, leading to reduced mitochondrial function and elevated ROS production, are thought to be a major contributor to the pathogenesis of ND. ER dysfunction has been implicated in ND in which protein misfolding is evidently causing ER stress. The consequences of ER stress ranges from an increase in ROS production to altered calcium efflux and proinflammatory signaling in glial cells. Both pathological pathways have links to ferroptotic cell death, which has been implicated to play an important role in ND. Pharmacological targeting of these pathological pathways may help alleviate or slow down neurodegeneration. I read the manuscript it is interesting and generally written well. Although, I felt it needs major modifications before considering it in this journal.

Please see the attached file (Comments and suggestions). 

Author Response

Reviewer #3

The paper entitled “CNS redox homeostasis and dysfunction in neurodegenerative  diseases “by Ryszard Gundars Goldsteins et al. is a review paper within the scope of Antioxidants.  The present review focuses on the major causes of CNS redox homeostasis imbalance comprising mitochondrial dysfunction and endoplasmic reticulum (ER) stress. Mitochondrial disturbances, leading to reduced mitochondrial function and elevated ROS production, are thought to be a major contributor to the pathogenesis of ND. ER dysfunction has been implicated in ND in which protein misfolding is evidently causing ER stress. The consequences of ER stress ranges from an increase in ROS production to altered calcium efflux and proinflammatory signaling in glial cells. Both pathological pathways have links to ferroptotic cell death, which has been implicated to play an important role in ND. Pharmacological targeting of these pathological pathways may help alleviate or slow down neurodegeneration. I read the manuscript it is interesting and generally written well. Although, I felt it needs major modifications before considering it in this journal.

Comments and suggestions

Introduction

  1. Introduction should end with a clear description of the article's scope, aims and structure. The main topics that will be discussed and the order in which these will be covered that is missing in your introduction. So I will recommend rewriting your introduction.

Response: Corresponding corrections introduced.

  1. Section 2 (Redox metabolism in mitochondria and production of ROS) although, your topic CNS redox homeostasis. Please check and confirm. I think your section title should be CNS redox homeostasis instead of Redox metabolism.

Response: Corresponding corrections introduced.

  1. In introduction CNS and ATP if you first time abbreviate with short form.

Response: Corresponding corrections introduced.

  1. Line 60, add following citation with reference 7. https://doi.org/10.3390/ijms222413313;

https://doi.org/10.1016/j.exger.2021.111352.

Response: The references added.

  1. Line 90-96, Alzheimer’s disease (AD) is the most prevalent neurodegenerative disease with a multi factorial origin. The disease affects especially the cerebral cortex and hippocampus. The disease’s hallmarks are recognized as aggregates of beta-amyloid (Aβ) and tau proteins that cause the deposition of amyloid plaques and neurofibrillary tangles, respectively. Aβ originates from the amyloid precursor protein (APP) that can undergo cleavage at different sites either by the non- amyloidogenic or amyloidogenic pathway, while tau aggregation is linked to its hyperphosphorylation. This all sentences have written without any references.

Response: The references added.

  1. Line 109 116, PGC-1α & BACE1 if you use first time introduce it.

Response: Corresponding corrections introduced.

  1. Line 129-133, Parkinson’s disease (PD) is the second most frequent neurodegenerative disorder with multifactorial background. At the histopathological level, it is characterized by the presence of intracellular lesions named Lewy bodies and by exacerbated cell death of dopaminergic

neurons. Similar to AD, PD can arise due to genetic causes (familial or heritable) or occur sporadically, this all sentences have written without references. Please check and confirm.

Response: The references added.

  1. Line 174-175, with reference 40 add following citation. https://doi.org/10.3390/pr9020308; https://doi.org/10.2174/1871527320666210218084444

Response: The references added.

  1. Line 216, pluripotent stem cells (iPSC), if you abbreviated before use short form only.

Response: Corresponding corrections introduced.

  1. Line 220-222, Oksanen et al have demonstrated that increased inflammation phenotype of presenilin-1 mutant iPSC-derived astrocytes is associated with increase calcium leakage from the ER, which is known to be inducible by ER stress [54]. Please check it has written correctly.

Response: Corresponding corrections introduced.

  1. Line 234, αSyn by autophagy, Please write it consistently throughout of the manuscript.

Response: Corresponding corrections introduced.

  1. Line 242-243, the role of ER stress and UPR in ALS is widely studied by utilizing various disease models, such as ALS iPSC lines, animal models and post-mortem ALS samples [4]. Check this sentence written correctly.

Response: Corresponding corrections introduced.

  1. Line unfolded protein response (UPR), if you abbreviated before then write only short form.

Response: Corresponding corrections introduced.

  1. Line 337, 338, 342, 350, 351, 363, 364,367,369, 371-375,377,378 αSyn Please write it

consistently throughout of the manuscript.

Response: Corresponding corrections introduced.

  1. Line 409, hydrogen peroxide (H2O2). Please check before you abbreviated it or not. It should be abbreviated with short form from first time then write short form throughout of the manuscript.

Response: Corresponding corrections introduced.

  1. Line 408-411, In the Fenton reaction Fe2+ and 408 hydrogen peroxide (H2O2) yields ferric iron (Fe3+) and hydroxyl radical (HO•). Next hydroxyl radical reacts with H2O2 producing superoxide (O2 - ). In the following Haber-Weiss 410 reaction superoxide reacts again with H2O2 and hydroxyl radical (HO•) and hydroxyl anion (-OH) are produced. Please make it consistent.

Response: Corresponding corrections introduced.

  1. Line 436-438, in concert with this, inhibition of ferroptosis 436 has been shown to decrease microglial activation and suppress the release of IL-6, IL-1β, and TNF-α [120]. If first time please abbreviate it.

Response: Corresponding corrections introduced.

  1. Line 459-460, In parallel, iron can promote tau hyperphosphorylation by induction of GSK3β 136]. Finally, a correlation between enhanced neuronal default mode network activity in carriers and cortical iron burden has been demonstrated, suggesting that interaction and Cdk5[ APOE4 between APOE

4 and iron may result in affected brain functions [137]. If first time please abbreviate it.

Response: Corresponding corrections introduced.

General comments:

Lot of typos and grammatical errors throughout of the manuscript that should be revised with help of native speaker or professional scientific writer.

Round 2

Reviewer 3 Report

The manuscript is improved a lot and now it is ready to be published in its current form in the journal.